# Occupational Risk for Post-Traumatic Stress Disorder and Trauma-Related Depression: A Systematic Review with Meta-Analysis

**DOI:** 10.3390/ijerph17249369

**Published:** 2020-12-14

**Authors:** Gabriela Petereit-Haack, Ulrich Bolm-Audorff, Karla Romero Starke, Andreas Seidler

**Affiliations:** 1Division of Occupational Health, Department of Occupational Safety and Environment, Regional Government of South Hesse, 65197 Wiesbaden, Germany; ulrich.bolm-audorff@rpda.hessen.de; 2Institute and Outpatient Clinic for Occupational and Social Medicine, University Medical Center Giessen, Justus-Liebig-University, 35392 Giessen, Germany; 3Institute and Policlinic of Occupational and Social Medicine (IPAS), Faculty of Medicine Carl Gustav Carus, Technische Universität Dresden, 01307 Dresden, Germany; karla.romero_starke@tu-dresden.de (K.R.S.); andreas.seidler@mailbox.tu-dresden.de (A.S.); 4Institute of Sociology, Faculty of Behavioral and Social Sciences, Chemnitz University of Technology, 09111 Chemnitz, Germany

**Keywords:** systematic review, occupational trauma, posttraumatic stress disorder, depression, occupational accident, occupational disease

## Abstract

There is evidence suggesting that occupational trauma leads to post-traumatic stress disorder (PTSD) and depression. However, there is a lack of high-quality reviews studying this association. We, therefore, conducted a systematic review with a meta-analysis to summarize the evidence of occupational trauma on PTSD and depression. After a database search on studies published between 1994 and 2018, we included 31 studies, of which only four had a low risk of bias. For soldiers exposed to wartime deployment, the pooled relative risk (RR) was 2.18 (95% CI 1.83–2.60) for PTSD and 1.15 (95% CI 1.06–1.25) for depression. For employees exposed to occupational trauma, there also was an increased risk for PTSD (RR = 3.18; 95% CI 1.76–5.76) and for depression (RR = 1.73; 95% CI 1.44–2.08). The overall quality of the evidence according to the Grading of Recommendations Assessment, Development, and Evaluation (GRADE) approach was moderate; the evidence was high only for the association between workers after exposure to trauma and development of PTSD. The study results indicate an increased risk of PTSD and depression in soldiers after participation in war and in employees after occupational trauma.

## 1. Introduction

In many professions, employees might experience traumatic events. For example, it is well known that rescue workers often experience an accident situation that is difficult to cope with when they arrive at accident sites. In a large number of professions (train drivers, soldiers, firefighters, police officers, paramedics, emergency doctors, journalists, photographers in war zones, prison staff, psychiatry staff, and others), there is a debate whether employees in such professions suffer from post-traumatic stress disorder (PTSD) or depression [1,2,3,4]. In Germany and several other EU countries, PTSD and depression can be recognized as an occupational accident [5,6]. Up to now, PTSD has only been recognized as an occupational disease in Denmark, while depression has been recognized as such in at least the Netherlands, Denmark, France, and Sweden [5,7]. In a systematic review, Utzon-Frank et al., 2014 [2] reported a 9.4% PTSD prevalence in firefighters zero to six months after an occupational trauma, and after six months or more, a PTSD prevalence of 12.1%. In their systematic review, Berger et al., 2012 [8] found a PTSD prevalence in firefighters of 7.3%. Sterud et al., 2006 [9], Berger et al., 2012 [8], and Hegg-Deloye et al., 2014 [10] described in their systematic reviews a PTSD prevalence of 4.0–25.5% in paramedics. All authors rated these prevalences to be increased compared to published data in the general population. However, the primary studies included in the mentioned reviews were not included in our systematic review because they were cross-sectional without a reference group.

Furthermore, in a meta-review (Bolm-Audorff et al., 2019 [11]), we found that all reviews [2,8,9,10] have considerable weaknesses. In this meta-review we assessed systematic reviews studying the relationship between occupational trauma, PTSD, and depression. One result of this metareview was that the quality of most systematic reviews according to AMSTAR (“Assessing the methodological quality of systematic reviews”) criteria [12,13,14,15] was low. Only about one-sixth of the systematic reviews had carried out their own quality assessment of the original studies included.

Based on our findings indicating a predominantly poor-quality basis in the existing reviews, we conducted our own systematic review with meta-analysis to investigate whether there is an increased occupational risk in particular occupational groups or as a result of specific occupational trauma for PTSD and depression. In particular, we conducted a systematic review and meta-analysis guided by the following research questions:Is there an increased risk of PTSD or depression among workers in specific occupations with frequent exposure to trauma (train drivers, soldiers, firefighters, police officers, paramedics, emergency doctors, war journalists or war photographers, prison staff, or psychiatric staff)? If so, how high is this risk?Is there an increased risk of PTSD or depression among employees who have been exposed to occupational trauma, violence, attacks, sexual harassment, or war? If so, how high is the risk?

## 2. Methods

A systematic electronic literature search was conducted in the Pubmed and Pilots databases on 20 September 2018 (Appendix A). Consideration was given to cohort studies, case-control studies and cross-sectional studies with a control group published between 1 January 1994–31 August 2018. The search started in 1994 due to the publication of the DSM (Diagnostic and Statistical Manual of Mental Disorders) IV criteria [16].

There were no language limitations for inclusion in the study, provided that the studies had an English or German abstract. The inclusion and exclusion criteria can be found in Appendix A. The review included all studies with employed persons of both sexes from the general population. We defined two exposure groups to differentiate between a specific occupational risk or a risk due to a specific trauma: exposure group 1 refers to special occupational groups that may have a higher risk of being exposed to occupational trauma (i.e., train drivers, soldiers, firefighters, police officers, paramedics, emergency doctors, journalists/photographers or similar in war zones, prison staff, psychiatric staff), while exposure group 2 refers to employees exposed to occupational trauma (trauma, violence, assault, sexual harassment in the workplace, wartime deployment).

Post-traumatic stress disorder, depressive disorder, and (other) affective disorders were accepted as outcomes. Only cohort studies, case-control studies or cross-sectional studies (all with a response ≥10%) were included in the study. An accepted comparison group for exposure group 1 was the general working population or occupational groups without evidence of increased exposure, and for exposure group 2, a group without exposure to occupational trauma (including groups not exposed to occupational trauma, e.g., soldiers without military service) was considered (for further details see Appendix A).

In addition, a hand search was performed by checking the reference lists of the included studies that did not refer to soldier studies. Additionally, a Google Scholar search was performed according to the “citation tracking factor.” To address the different occupational groups covered in our meta-review, the following studies were included in the “citation tracking” Google Scholar search: Aoki et al., 2012 [17] (journalists), Utzon-Frank et al., 2014 [2] (different professions), Bills et al., 2008 [18] and Sterud et al., 2006 [9] (disaster relief workers), and Clarner et al., 2015 [19] (train drivers). Qualitative studies, ecological studies, case reports, and experimental studies were excluded.

The titles and abstracts were independently reviewed by two reviewers (U.B.-A. and G.P.-H.) to determine whether the inclusion criteria were met. If no agreement could be reached, a third author (A.S.) was consulted and the study was discussed until a majority consensus was reached. The full texts of the studies included after the review of titles and abstracts were then reviewed by two independent evaluators (U.B.-A. and G.P.-H.). In the absence of agreement, the same procedure was followed as for the review of the titles and abstracts. The reasons for exclusion were recorded for each excluded study.

The data extraction was carried out independently by two reviewers (U.B.-A. and G.P.-H.), who discussed results with each other in case of discrepancies. If no agreement could be reached, the data extraction was discussed together with a third author (A.S.) until a majority was reached. The following information was collected from each included study: first author, year of publication, study region, study design, type of employees investigated, number of exposed and non-exposed persons, response (%), age, time period in which the study was conducted, duration of exposure, job tasks, instrument used for evaluation of PTSD or depression, research institution, funding sources, disease frequency, number of exposed cases and controls, effect estimate, confounders considered, and additional analyses done.

The risk of bias evaluation was carried out according to the study by Ijaz et al., 2013 [20] and was performed by two reviewers (U.B.-A. and G.P.-H.), independently of each other. When there was a disagreement, the study was discussed with a third author (A.S.) until a majority consensus was reached. Six major domains with important sources of bias were evaluated (study recruitment and follow-up, exposure definition and measurement, outcome source and validation, confounding and effect modification, assessment, quality of the statistical analysis, and chronology). Each domain could be judged as having a high risk of bias, a low risk of bias, or an unclear risk of bias. If a study was evaluated with a high or unclear risk of bias in one of the six major domains mentioned above, the study was evaluated as having a high risk of bias. In addition, the following minor domains of bias were recorded: blinding of the evaluators, study funding, and conflict of interest. However, these did not play a role in the overall evaluation of bias for the study.

The study results were summarized descriptively and in meta-analyses (K.R.S). Because odds ratios (ORs) tend to overestimate the relative risk when the prevalence of the outcome of interest is high, we converted the ORs to prevalence ratios (PRs) when the prevalence was higher than 10% using the formula proposed by Zhang and You 1998 [21]. When studies did not directly report PRs, this was manually done if the studies included the necessary information to do so. Meta-analyses were performed to estimate the pooled risk of occupational trauma on depression and PTSD. The meta-analysis was carried out if at least two primary studies which were comparable in terms of exposure and outcome were included. Excluded were studies that had no effect estimator or had no information on trauma. Due to the heterogeneity of the studies, the random effects model was used as the analysis method. The I^2^ value in general was given as a measure of heterogeneity, keeping in mind that the I^2^ statistic depends on the size of the studies included [20,21]. The occurrence of publication bias was determined using funnel plots and Egger’s tests. Stata Version 14.2 (StataCorp, College Station, TX, USA) was used for the meta-analysis.

We used the Grading of Recommendations Assessment, Development, and Evaluation (GRADE) approach for grading the quality of the total body of evidence [22], following the example of Hulshof et al., 2019 [23], with modifications (Romero Starke et al., 2019 [24,25]). We used the following levels of quality: high, moderate, and low. An initial “high” level would indicate having randomized studies. If only observational studies were included, then the starting level would be set to “moderate.” The quality of evidence was downgraded based on five factors: study limitations (high risk of bias), indirectness, inconsistency, imprecision (range of the CI of studies > 2.0), and publication bias (yes or unclear). Study findings with large effect sizes (an effect estimate > 2.0), a dose-response relationship, or the presence of residual confounding (which would increase confidence in the association), resulted in an upgrade of the quality of evidence. If an effect size larger than 5.0 was present, the quality of evidence was upgraded twice.

The study design was published a priori in PROSPERO (Prospero registration number: CRD42019122774, Petereit-Haack et al., 2019 [26]). The review meets all criteria of AMSTAR 2 [15].

## 3. Results

### 3.1. Search Results

The Prisma flow chart (Figure 1) shows the results of the literature search. After exclusion of duplicates, we reviewed 12,321 and excluded 12,180 studies based on the title and abstract. After the full-text review, we excluded 110 more studies, yielding 31 included publications [1,27,28,29,30,31,32,33,34,35,36,37,38,39,40,41,42,43,44,45,46,47,48,49,50,51,52,53,54,55]. Appendix A shows the studies excluded based on the full-text review. The reason for exclusion of 25 studies in the full text review was due to the study design. This was mainly due to a lack of response information or due to a response below 10%. Sixty-five studies were excluded either due to the absence of a comparison group or due to the use of an unsuitable comparison which had a similar exposure to that of the study group. In 16 studies, the exposure was not pertinent, and four studies were excluded based on the outcome (Appendix A).

### 3.2. Study Characteristics

#### 3.2.1. Study Design and Country of Study

The inclusion criteria were met by 31 studies with the following designs: 28 cross-sectional studies, 2 cohort studies [40,47], and 1 matched case-control study [53]. Twelve studies originated in Europe [27,28,31,32,39,40,42,49,52,53,54,56], nine studies were from North America [35,37,38,41,45,46,47,48,50], four studies were from Asia [33,34,51,55], two studies originated from South America [29,43], two studies were from Australia [36,44], one study was from Israel [57], and one study included data from around the world [30]. The studies are listed in Appendix A.

#### 3.2.2. Outcomes Studied

Study Population

Ten studies were conducted on soldiers and/or veterans after wartime deployment [28,31,32,35,37,38,41,46,47,50]; seven studies were done on health care workers [36,39,44,45,49,55,57]; three studies included police officers or firefighters [27,32,52]; three studies investigated train drivers or subway train drivers [33,34,56]; and two studies were done on journalists or war journalists [29,30]. Thirteen studies investigated PTSD [28,31,32,37,39,41,44,45,46,47,54], 10 studies investigated depression [36,40,42,43,49,51,52,53], and another 10 investigated both outcomes [29,30,33,34,35,38,48,50,56,57]. The evaluation of the outcomes varied among the studies. For the outcome PTSD, 16 different instruments were used: IES-R (Impact of Event Scale Revisited) [29,30,45,57], MINI (International Psychiatric Interview) [41,56], PTSD Checklist (PCL-L) [31,35,44,50], Self-rating inventory for post-traumatic stress disorder [32], AUDADIS-5 (Alcohol Use Disorder and Associated Disabilities Interview Schedule DSM-5 Version) [37], CHAMPS (Inpatient or outpatient ICD-9 diagnosis of PTSD according to the archival system for medical personnel) [38], PSS-SR (Self-report Post-traumatic Stress Disorder Symptom Scale, DSM-III) [39], Trauma Assessment for Adults Questionnaire Clinician Administered PTSD Scale [41,45], and the US army data system about treatment for PTSD [47]. For the evaluation of depression, 15 different instruments were used: MINI (International Psychiatric Interview) [50,56,57], HADS (Hospital Anxiety and Depression Scale) [27,42,49], BDI II (Beck depression inventory II) [28,29,30], GHQ 28 (depression subscale of General Health Questionnaire) [29,30,56], SCID (Clinical interview for axis I DSM IV disorders) [30], CIDI (WHO Composite International Diagnostic Interview) [33,34,48], Depression Scale of the Patient Health Questionnaire, DSM IV [35], CHAMPS (Inpatient or outpatient ICD-9-diagnosis of PTSD according the archival medical personnel system) [38], SF-26 (5-item mental health inventory of the 36-item short-form) [40], register of medicinal product statistics [40], PHQ-9 (major depressive disorder, MDD, Diagnostic and Statistical Manual of Mental Disorders, 4th Edition, DSM-IV) [34], CES-D (Japanese version) [51], SCL-90-R (System Checklist Revised) [52], ICD (Clinical Psychiatric Diagnoses) [53], and the Self-rating depression scale (SDS) [55]. For details, see Appendix A.

Exposure

In 17 studies, the exposure was defined as having had a trauma at the workplace [28,29,32,33,34,36,39,40,42,43,44,45,48,51,52,53,56,57]. Fourteen studies investigated an occupational group [27,30,31,34,35,37,38,41,46,47,49,50,54,55].

Comparison Group

In most studies, unexposed persons of the same occupational group were used as the comparison group, whereas in six studies the control persons came from the general population [27,40,42,43,49,53]. The response was between 11.9% and 100%. Further details on the response, age, and sex distribution can be found in Appendix A.

### 3.3. Risk of Bias Evaluation

In total, four studies showed a low risk of bias [38,40,47,53], and all remaining 27 studies showed a high overall risk of bias. Of the 10 studies which investigated PTSD and depression as outcomes, only one study had a low risk of bias [38]. Two of the 10 studies that evaluated only depression as their outcome were evaluated as having a low risk of bias [40,53]. Similarly, only 1 study [47] of the 13 studies evaluating PTSD as an outcome had a low risk of bias. All 27 cross-sectional studies were classified as having a high risk of bias because of the study design (Table 1).

Five studies had a high risk of bias due to the recruitment procedure [27,39,44,45,49]. Two studies had a high risk of bias due to the exposure [27,28]; 1 study because of the outcome [46]; 11 studies due to confounders [28,29,36,37,39,44,45,46,51,54,55]; and 12 studies due to the analysis methods [27,28,29,30,39,44,45,46,49,54,56,57]. Results with regard to the minor domains are shown in Table 1. The details of the risk of bias analysis for each of the 31 studies can be provided by the authors on request.

### 3.4. Study Results

The results of the individual studies are shown in Appendix A.

A significant association between nurses exposed to violence at the workplace and depression was described by Zhao et al., 2018 [55]. This study could not be included in the meta-analysis because no effect estimator was provided. In the population-based study from France, Niedhammer et al., 2015 [42] described a statistically significant association between physical violence or sexual assault and the risk of depression in men, but not in women. This study could also not be included in the meta-analysis because an effect estimator was not provided.

Luce et al., 2002 [39] described more pronounced levels of PTSD in a survey of health care workers who treated victims of the 1998 Omagh bombing in Northern Ireland compared to health care workers who had no contact with these victims. Opie et al., 2010 [44] described a significant correlation between the extent of sexual harassment experienced by nurses and the severity of PTSD in very lonely areas of Australia. Park, 2011 [45] found an increased prevalence of PTSD in nurses involved in the management of Hurricane Katrina.

We found the following results for occupational groups with a high risk of trauma:

Firefighters: According to the study by Huizink et al., 2006 [32], firefighters who had to extinguish a fire after a plane crash did not have an increased risk of posttraumatic stress disorder compared to firefighters without this exposure (OR = 1.1, 95% CI 0.4–3.7). Van der Velden et al., 2013 [52] described a prevalence of pronounced depressive symptoms in 13.0% of 123 firefighters examined. The prevalence of depression was not significantly increased compared to the comparison group of 144 police officers without specific traumas, of whom 11.8% had such symptoms.

Police officers: In a cross-sectional study of 3272 Norwegian police officers, Berg et al., 2006 [27] reported no difference in the expression of depressive symptoms in this group compared to the control group from the general population after age stratification. Specific traumas in police officers, such as the involvement in a shoot-out or experiences of violence at work, were not described in their study. In their cross-sectional study of 834 police officers, Huizink et al. [32] found that police officers involved in coping with a plane crash had 2.8 times the risk of depression (95% CI 1.5–5.0), compared to police officers who had nothing to do with the crash. Van der Velden et al., 2013 [52] reported in a cross-sectional study that police officers who were professionally involved in the handling of the fire disaster in Enschede in 2000, had a significantly lower risk of developing depression compared to police officers who were not involved in this fire disaster.

Paramedics: In a cross-sectional study of 1180 paramedics in Norway, Sterud et al., 2008 [49], described the prevalence of depression in men as 8.0% compared to that of the general resident population (7.0%). In contrast, the prevalence of depression was lower among female paramedics compared to the unexposed comparison group (4.6 vs. 6.7%). Since no information on trauma was provided in the study, this study was not included in the meta-analysis.

War journalists: In their cross-sectional study of 140 war journalists, Feinstein et al., 2002 [30] described a significantly more pronounced PTSD and depression symptomatology in war journalists compared to journalists without war coverage. Furthermore, in a cross-sectional study of 87 Mexican journalists threatened by drug cartels, Feinstein et al., 2012 [29] did not find a significant increase in PTSD and depression in such journalists, compared to 17 journalists who were not exposed to such threats. A meta-analysis was not possible because neither study provided effect estimates on the risk of disease.

Subway drivers: Two studies considered subway drivers: Kim et al., 2013 [33] and Kim et al., 2014 [34]. The PTSD risk for subway drivers was significantly increased in these studies: 11.70; 95% CI 1.90–225.80 for Kim et al., 2013 [33] and 1.54; 95% CI 0.52–4.55 for Kim et al., 2014 [34]. The risk of depression for subway drivers also showed an increased risk of depression (RR = 2.60; 95% CI 0.70–9.40) for Kim et al., 2013 [33] and (RR = 1.99; 95% CI 0.72–5.53) for Kim et al., 2014 [34]. However, the values were not statistically significant since both studies only had a small number of exposed and unexposed subway drivers with depression.

Soldiers: In a sub-sample of their cohort (*n* = 326), Hotopf et al., 2006 [31] could not show an association between soldiers after war deployment and PTSD (OR = 0.61; 95% CI 0.39–0.95). In a further study of 296 participants, there was a statistically significant increased risk of PTSD in soldiers after war deployment (Kline et al., 2010 [35]). In another study (Lehavot et al., 2018 [37]) PTSD increased by more than 2.65 times in women (OR = 2.65; 95% CI 0.83–8.41) and by more than five times in men (OR = 5.05; 95% CI 3.60–7.10). A cohort study investigated both PTSD and depression in army soldiers and marines (Levin-Rector et al., 2018 [38]), and also found an increased risk in both outcomes for both soldier groups 11 (PTSD in army soldiers: HR = 1.74; 95% 1.71–1.76, PTSD in marines: HR = 2.04; 95% CI 1.93–2.15, depression in army soldiers: HR = 1.11; 95% CI 1.09–1.12, depression in marines: HR = 1.12; 95% CI 1.07–1.17). In the studies by Wittchen et al., 2012 [54], Thomas et al., 2017 [50], Proctor et al., 1998 [46], and Magruder et al., 2005 [41], the number of cases with PTSD in exposed and non-exposed soldiers was well below 100 persons. The study with the highest number of cases of PTSD (*n* = 1844) showed a strongly increased PTSD risk for female soldiers exposed to sexual assault (OR = 6.3; 95% CI 5.7–6.9) (Rosellini et al., 2017 [47]).

### 3.5. Meta-Analysis

A total of 22 studies were included in the various meta-analyses. Excluded were eight studies [27,29,30,39,42,46,55,57] due to the lack of an effect estimator and one study [49] due to lack of information on a specific trauma in police staff. Nine of the included studies were concerned with soldiers after war deployment and PTSD (Figure 2); four studies dealt with depression in soldiers after war deployment (Figure 3); six studies examined PTSD in workers after exposure to occupational trauma (Figure 4); and nine other studies examined depression in relation to occupational trauma (Figure 5). Funnel plots and sensitivity analyses are shown in Appendix A.

#### 3.5.1. PTSD in Soldiers after War Deployment

The meta-analysis included nine soldier studies with PTSD [28,31,35,37,38,41,48,50,54]. The pooled effect resulted in a roughly doubled risk of PTSD in soldiers after war deployment (RR = 2.18; 95% CI 1.83–2.60). The results are shown in Figure 2. When considering the soldiers’ studies according their risk of bias, the studies with a higher risk of bias had a higher effect estimate (RR = 2.46; 95% CI 1.41–4.29) than the studies with a low bias risk (RR = 1.88; 95% CI 1.61–2.20) (see Figure 3).

Five different instruments (CIDI, PCL, AUDADIS, register data, and CAPS) were used to measure PTSD in the studies. The pooled relative risk for PTSD in soldiers after war deployment varied with the instrument used: it was 2.13 (95% CI 1.32–3.44) when CIDI was used and 1.47 (95% CI 0.61–3.55) when the outcome was measured with PCL (see Appendix A).

Overall, there was no evidence of publication bias (Appendix A).

#### 3.5.2. Depression in Soldiers after War Deployment

Four studies were included in the meta-analysis for depression in soldiers after war deployment [35,38,48,50]. The pooled risk estimate was statistically significant elevation at 1.15 (95% CI 1.06–1.25, Figure 4.) Studies with a high risk of bias showed a higher pooled relative risk (RR = 1.60; 95% CI 0.93–2.74) than studies with a low risk of bias (1.11; 95% CI 1.10–1.13), see Appendix A. There was no indication of publication bias (Appendix A).

#### 3.5.3. PTSD in Workers Exposed to Occupational Trauma

The PTSD risk for employees exposed to occupational trauma was strongly increased (RR = 3.18; 95% CI 1.76–5.76, Figure 5). Six studies were included in this meta-analysis [32,33,34,45,47,56]. Studies with a high risk of bias showed a lower pooled relative risk (RR = 2.45; 95% CI 1.36–4.39) than the only study [47] with a low risk of bias (RR = 5.20; 95% CI 4.82–5.61), see Appendix A. Overall, there was no indication of publication bias (Appendix A).

#### 3.5.4. Depression in Workers Exposed to Occupational Trauma

The pooled relative risk estimate of the nine occupational trauma studies included in the meta-analysis [16,33,34,36,40,43,51,52,53,56] showed a 77% increased risk of depression for employees with work-related trauma (RR = 1.77; 95% CI 1.45–2.15, Figure 6.). Studies with a high risk of bias had a higher effect estimate (RR = 2.10; 95% CI 1.90–2.33) than the two studies with a low risk of bias [40,53], RR= 1.44; 95% CI 1.29–1.61), shown in Appendix A. There was no indication of publication bias according to the funnel plot’s and Egger’s test (Appendix A).

### 3.6. Quality of Evidence Assessment (GRADE)

For the quality of evidence assessment, we started with a “moderate” quality level, because only observational studies were included (Table 2).

#### 3.6.1. PTSD in Soldiers after War Deployment

The initial quality level was neither downgraded nor upgraded for any of the categories. The I^2^ was high for the overall effect and for the effect of the low risk of bias studies. However, the low risk of bias studies were large studies with high precision and the confidence intervals overlap. It has already been described that I^2^ is not a good measure of heterogeneity in large studies [58,59], and therefore, it was decided that there was no indication of inconsistency. Although the pooled effect estimate was greater than 2.0, the high and low risk of bias studies differed significantly from each other (high risk RR = 2.46; 95% CI 1.41–4.29; low risk RR = 1.88; 95% CI 1.61–2.20), and the low risk of bias studies had an effect lower than 2.0. Therefore, we decided not to upgrade due to this category. We finally judged that the quality of evidence for increased risk of PTSD in soldiers after war deployment to be “moderate” (see Table 2).

#### 3.6.2. Depression in Soldiers after War Deployment

All categories remained unchanged after the initial grading, and therefore, the quality of evidence for increased risk of depression in soldiers after war deployment was assessed to be “moderate.”

#### 3.6.3. Depression in Workers Exposed to Trauma

The initial quality level was neither downgraded nor upgraded due to any of the categories, and therefore, the quality of evidence for increased risk of depression in workers exposed to trauma was judged to be “moderate.”

#### 3.6.4. PTSD in Workers Exposed to Trauma

The quality level was downgraded once because of high imprecision, and it was upgraded twice to reflect the high effect of the pooled estimate (>5.0). Although the pooled effect estimate was 3.18, which would normally cause one upgrade, the high and low risk of bias studies had pooled effects which differed significantly from each other (low risk RR = 5.20; 95% CI 4.82–5.61; high risk RR = 2.45; 95% CI 1.36–4.39). The low risk of bias study had an effect greater than 5.0, and thus, the quality level was upgraded twice. All other categories remained unchanged. The overall quality of evidence for increased risk of depression in workers exposed to trauma was “high.”

## 4. Discussion

This systematic review with meta-analysis finds considerably increased PTSD risks and slightly increased depression risks among soldiers after war deployment. The significantly decreased PTSD risk in the study of Hotopf et al., 2006 [31] is possibly influenced by a “healthy soldier” effect. According to two studies, subway drivers exposed to a person under train (PUT) accident are prone to considerably elevated PTSD risks and potentially elevated depression risks (lacking statistical significance). For the other examined occupational groups with potential traumatic exposure to frequent trauma (firefighters, police officers, paramedics, war journalists), no consistent PTSD or depression risk increases could be observed. We could not identify studies on emergency doctors, prison staff, or psychiatric staff that fulfilled the inclusion criteria.

In workers exposed to trauma, there was a 3-fold increase in PTDS risk, while a 1.7-fold increased risk of depression was observed.

### 4.1. PTSD Risks due to Occupational Trauma

PTSD risks were found to be considerably increased among soldiers after war deployment. War deployment is characterized by frequent exposure to traumatic situations. However, the respective studies included in our systematic review did not differentiate between subjects with and without exposure to specific traumatic situations, e.g., serious injury of the soldier or confrontation with dead comrades. Therefore, the PTSD risk of soldiers after war deployment might be considerably underestimated. For other occupations with potential for exposure to traumatic events (firefighters, police officers, paramedics, war journalists)—albeit presumably being more rarely exposed compared to war-deployed soldiers—we did not observe consistently elevated PTSD risks. However, when we looked at persons who were occupationally exposed to traumatic events, PTSD risk was clearly elevated (RR = 3.18; 95% CI 1.76–5.76, Figure 5). We, thus, infer from our results that PTSD risks cannot be adequately reflected by using the occupational group as a proxy variable for occupational traumas.

Most of the included studies did not differentiate between exposures to one or more traumatic events. Kim et al., 2013 [33] reported a prevalence ratio for one-year PTDS after one PUT(person under train) accident of 13.4 (95% CI 1.9–265.3) and of 8.7 (95% CI 0.7–201.7) after ≥2 PUT accidents (Appendix A). The risk estimates are wide due to the low number of PTSD cases. Furthermore, there might be a differential selection after the first PUT incident if subway workers leave the profession due to the PUT experience. PTSD might not only develop as a consequence of a single traumatic event, but it might also result after exposure to multiple events over time. As Priebe et al., 2018 [60] suggest, there may be repeated micro-aggressions against police officers or psychiatric staff, and paramedics and emergency doctors might be exposed to multiple experiences of serious accidents, possibly lacking a single worst incident. Accordingly, the DSM-IV (American Psychiatric Association 2000 [61]) Criterion A “has been exposed to a traumatic event” has been modified in the DSM-5 (American Psychiatric Association 2013 [62]) to “traumatic event(s),” therefore comprising multiple events or a recurring exposure. Stein and colleagues (2016) [63] pointed out that subjects exposed to multiple traumas might develop different symptoms by each trauma, which (only) in combination might fulfil the criteria for PTSD; accordingly, the authors speak of cumulative trauma. In a multi-national survey (Karam et al., 2014 [64]), about 20% of the subjects with 12-month PTSD reported symptoms that were associated with more than one traumatic event. While PTSD due to a single occupational trauma would be regarded as an occupational accident in most countries, PTSD due to multiple traumatic events could in principle be regarded as an occupational disease (depending on country-specific legal understandings).

### 4.2. Depression Risks due to Occupational Trauma

Among soldiers after war deployment, depression risks were slightly elevated, but they were nonetheless statistically significant. The slight risk increase of only 15% might at least partly be explained by a healthy worker effect (healthy hire effect). Moreover, the small magnitude of effect might be due to risk “dilution,” as not all deployed soldiers are expected to have been exposed to traumatic events.

According to our meta-analysis, occupational traumatic events were clearly associated with depressive symptoms (pooled relative risk of 1.77; Figure 6). Even though our GRADE assessment resulted in an overall “moderate” certainty level, we must emphasize that the high-quality studies still provided a statistically significant increased effect (44%) of occupational trauma on depression. Most of the applied diagnostic instruments (BDI_R; SF-36; PHQ-9; CES-D; GHQ; SCL-90-R) cannot clearly differentiate trauma-related depressions from adjustment disorders (sometimes also called situational depressions; https://icd.codes/icd10cm/F432), which might also result from traumatic events. This is because symptoms of adjustment disorders are similar to those of affective disorders (e.g., depression; anxiety; conduct or emotional disturbance (Gradus 2017 [65])). Accordingly, three of seven ICD-10 adjustment disorder subtypes include “depressive reaction” in their title (Maercker and Lorenz 2018 [66]). In principle, adjustment disorders resolve in the course of six months [65]. However, in some cases (if the stressor persists for a longer duration or in case of a prolonged depressive reaction; ICD-10 F43.1), adjustment disorders can remain for a longer time, which complicates the differential diagnosis between adjustment disorder and trauma-related depression. Unfortunately, in all studies included in our systematic review information about the time lag between traumatic event and depression diagnosis is lacking. A clear diagnostic assignment of the depressive symptoms is, therefore, not possible.

Several studies reveal high psychiatric comorbidities after trauma exposure (Maercker and Hecker 2016 [67]). Particularly the co-occurrence of PTSD and depression has been consistently demonstrated in the epidemiologic literature [65]. According to a large cohort study among US active duty service members, PTSD occurred most frequently together with depressive disorders (49.0%), adjustment disorders (37.0%), generalized anxiety disorders (36.1%), and alcohol use disorders (26.9%) [68]. In a recently published study among 223 US veterans with combat-related PTSD, Goetter et al. (2020) [69] reported that about 70% of the veterans had comorbid major depressions. Occupational traumas might, therefore, play a role in the development of not only of PTSD, but also depression and other psychiatric diseases, frequently occurring simultaneously in the same employees. However, the temporal relationship and the etiological directionality are not always clear from the currently published studies. Future research on the temporal relationship between occupational traumas and depression as well as PTSD should conduct several follow-ups, taking into account the time course of psychiatric symptoms including PTSD, adjustment disorders, and depression (at best including a clinical diagnosis in addition to psychological assessments).

### 4.3. Strengths and Limitations

The main strengths of this review were the systematic literature search conducted using a comprehensive search string in two databases (Pubmed and Pilots), the inclusion of studies in all languages, the evaluation of titles, abstracts, and full texts by two independent researchers, and the dual assessment of the study quality, finding a consensus to determine the final quality level of the included studies. The formal risk of bias assessment was integrated into our analysis and conclusions. Studies using a convenience sample or with either no reported response or a very low response were excluded to minimize selection bias.

Unlike previous reviews, this review included only studies with an unexposed comparison group, and therefore, the respective measure of effect due to occupational trauma could be pooled in a meta-analysis. However, most studies compared workers of the same occupation with and without a specific trauma, or with and without war deployment [27,28,29,30,31,32,33,34,35,36,37,38,39,41,44,45,46,47,49,50,52,54,55,56,57]. For instance, Huizink et al., 2006 [32] performed a cross sectional study in firefighters and police officers involved in the handling of an aircraft crash and compared them with firefighters and police officers without involvement in the crash. This comparison might underestimate the true disease risk because firefighters and police officers not involved in the air crash are possibly more exposed to occupational trauma compared to the general working population (i.e., in the rescue of injured or dead victims of normal fires or in the confrontation with aggressive persons during normal police work). One study (van der Velden et al., 2013 [52]) compared bank employees after robbery with police officers without massive trauma. This comparison, again, may possibly underestimate the disease risk because of the exposure of police officers to occupational trauma during their normal work. Only a few studies used the general population as a comparison group [27,40,42,43,49,53].

One of the limitations of the present review is that the types of occupational trauma associated with an increased risk of PTSD and depression were very different (Figure 5 and Figure 6). Our meta-analysis included studies of subway and train drivers after PUT experiences [33,34,56]; firefighters after airplane crashes [32]; nurses who had weathered Hurricane Katrina [45] or who had experienced particularly stressful events in hospital, such as the death or serious injury of a child [36]; female soldiers with a condition following rape or other sexual trauma [47]; police officers after deployment in a plane crash or fire disaster [32,52]; bank employees after a robbery [52]; and workers exposed to violence at work [40,43,51,53]. Only for the trauma of soldiers participating in war were there sufficient studies to conduct a separate meta-analysis.

In the studies, the PTSD and depression outcomes were assessed exclusively by questionnaires. In some singular studies, a specialist diagnosis was also performed. Moreover, the included studies diagnosed PTSD with 10 different instruments, and it is unknown whether the measured effects are comparable between the instruments. As an example, the pooled relative risk for PTSD for soldiers after war deployment was 2.13 (95% CI 1.32–3.44) when the CIDI instrument used, whereas it was 1.47 (95% CI 0.61–3.55) when the PCL instrument was used (Appendix A). Depression was also measured using various instruments between the studies—15 instruments in total.

## 5. Conclusions

Our results show a doubling of the risk for PTSD and a significantly increased risk for depression in soldiers exposed to war deployment. Furthermore, there is a tripled risk of PTSD and almost a doubled risk for depression on workers exposed to occupational trauma. These findings have an important impact on public health because of the prevalence of traumatic events in particular occupational groups, along with a considerable frequency of PTSD, depression, and other affective disorders in the general population. Recognizing that members of some specific occupational groups (such as police officers, firefighters, and nurses) might not be exposed to a “single worst incident,” but rather might experience multiple distressing events in the workplace, research should focus on quantifying the impact of these repeated workplace experiences on PTSD, depression, and other related disorders, using an appropriate comparison group.

## Figures and Tables

**Figure 1 ijerph-17-09369-f001:**
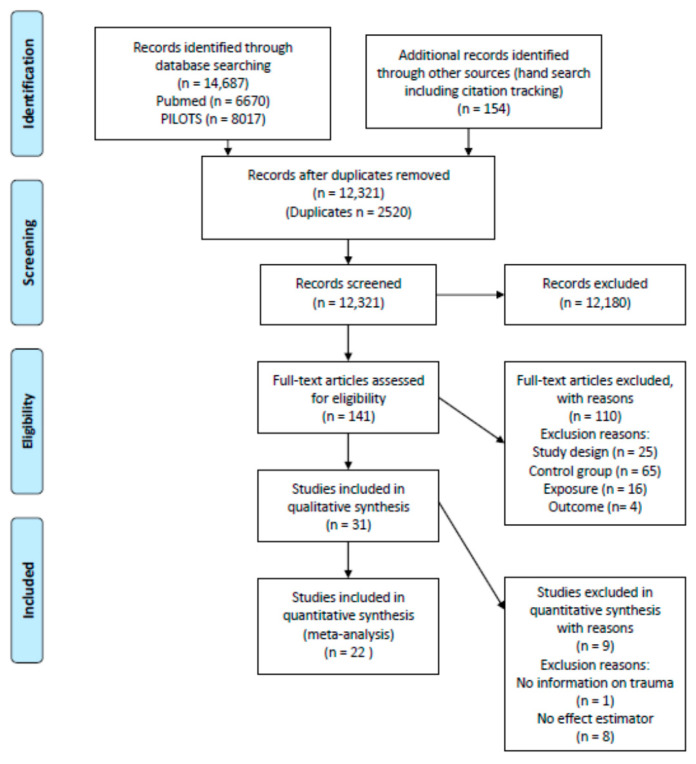
Flow diagram of inclusions and exclusions.

**Figure 2 ijerph-17-09369-f002:**
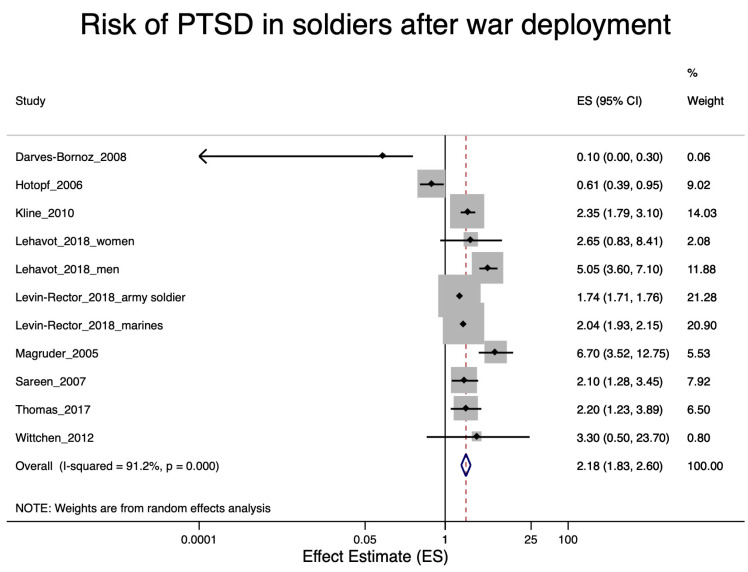
Forest plot of soldiers after war deployment and the estimates of effect size (ES) for post-traumatic stress disorder (PTSD).

**Figure 3 ijerph-17-09369-f003:**
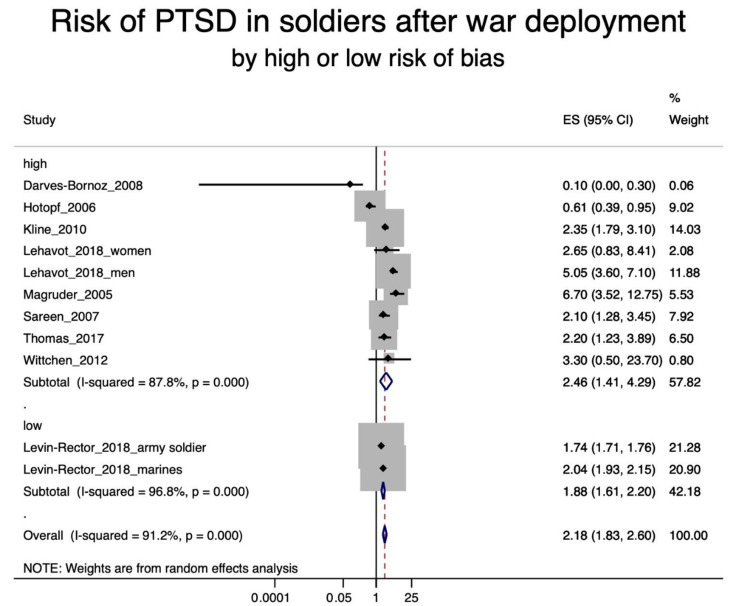
Forest plot of soldiers after war deployment and the estimates of effect size (ES) for post-traumatic stress disorder (PTSD) by high or low risk of bias.

**Figure 4 ijerph-17-09369-f004:**
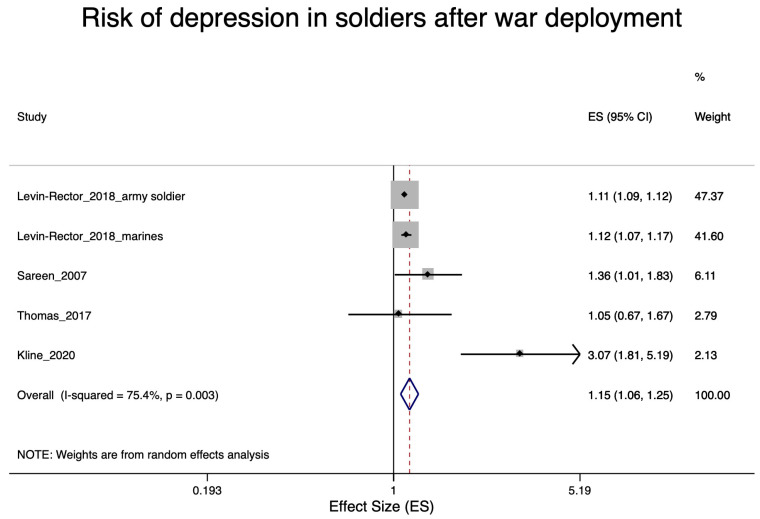
Forest plot of soldiers after war deployment and the estimates of effect size (ES) for depression.

**Figure 5 ijerph-17-09369-f005:**
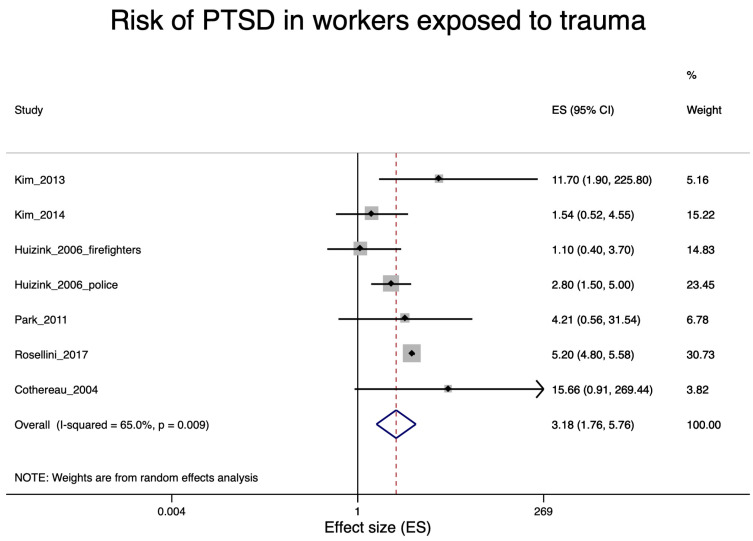
Forest plot of workers exposed to occupational trauma and the estimates of effect size (ES) for post-traumatic stress disorder (PTSD). Note: unadjusted prevalence ratio (PR) for Park et al., 2011 [45] and Cothereau et al., 2004 [56]; Cothereau et al., 2004 [56] was included by adding a 0.5 case to all categories.

**Figure 6 ijerph-17-09369-f006:**
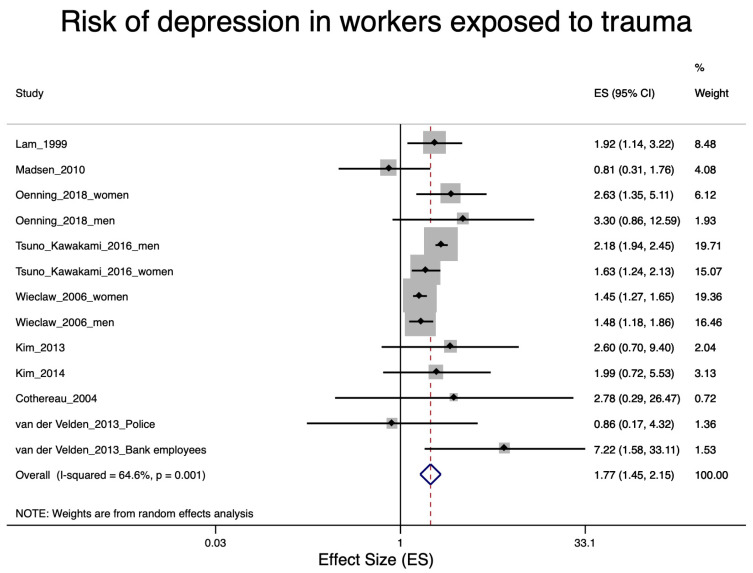
Forest plot of workers exposed to occupational trauma and the estimates of effect size (ES) for depression. Note: unadjusted PR for Cothereau et al., 2004 [56].

**Table 1 ijerph-17-09369-t001:** Risk of Bias.

			Major Domains					Minor Domains				Overall
No	Study ID	Outcome	1. Recruitment Procedure & Follow-up (in Cohort Studies)	2. Exposure Definition and Measurement	3. Outcome. Source and Validation	4. Confounding and Effect Modification	5. Analysis Method: Methods to Reduce Research Specific Bias	6. Chronology	7 Blinding	8. Funding	9. Conflict of Interest	
1	Berg et al. 2006 [27]	Depression	0	0	2	2	0	0	1	2	2	0
2	Darves-Bornoz et al. 2008 [28]	PTSD	2	0	2	0	0	0	1	2	1	0
3	Feinstein et al. 2002 [29]	PTSD, Depression	2	2	2	2	0	0	1	2	1	0
4	Feinstein 2012 [30]	PTSD, Depression	2	2	2	0	0	0	1	1	1	0
5	Hotopf et al. 2006 [31]	PTSD	2	2	2	2	2	0	1	2	0	0
6	Huizink et al 2006 [32]	PTSD	2	2	2	2	2	0	1	1	1	0
7	Kim et al. 2013 [33]	PTSD, Depression	2	2	2	2	2	0	1	2	1	0
8	Kim et al. 2014 [34]	PTSD, Depression	2	2	2	2	2	0	1	2	1	0
9	Kline et al. 2010 [35]	PTSD, Depression	2	2	2	2	2	0	1	2	0	0
10	Lam et al. 1999 [36]	Depression	2	2	2	0	2	0	1	1	1	0
11	Lehavot et al. 2018 [37]	PTSD	2	2	2	0	2	0	1	2	2	0
12	Levin-Rector et al. 2018 [38]	PTSD, Depression	2	2	2	2	2	2	2	2	0	2
13	Luce et al. 2002 [39]	PTSD	0	3	3	0	0	0	1	1	1	0
14	Madsen et al. 2010 [40]	Depression	2	2	2	2	2	2	1	3	1	2
15	Magruder et al. 2005 [41]	PTSD	2	2	2	2	2	0	1	2	1	0
16	Niedhammer et al. 2015 [42]	Depression	2	2	2	2	2	0	1	2	2	0
17	Oenning et al. 2018 [43]	Depression	2	2	2	2	2	0	1	2	2	0
18	Opie et al. 2010 [44]	PTSD	0	2	2	0	0	0	1	1	1	0
19	Park et al. 2011 [45]	PTSD	0	2	2	0	0	0	1	1	1	0
20	Proctor et al. 1998 [46]	PTSD	2	2	0	0	0	0	1	2	1	0
21	Rosellini et al. 2017 [47]	PTSD	2	2	2	2	2	2	2	2	1	2
22	Sareen et al. 2007 [48]	PTSD Depression	2	2	2	2	2	0	0	2	2	0
23	Sterud et al. 2008 [49]	Depression	0	2	2	2	0	0	1	2	2	0
24	Thomas et al. 2017 [50]	PTSD Depression	2	2	2	2	2	0	1	2	2	0
25	Tsuno et al. 2016 [51]	Depression	2	2	2	0	2	0	1	2	2	0
26	van der Velden et al. 2013 [52]	Depression	2	2	2	2	2	0	2	1	2	0
27	Wieclaw et al. 2006 [53]	Depression	2	2	2	2	2	2	2	2	2	2
28	Wittchen et al. 2012 [54]	PTSD	2	2	2	0	0	0	1	2	2	0
29	Zhao et al. 2018 [55]	Depres-sion	2	2	2	0	2	0	1	2	2	0
30	Cothereau et al. 2003 [56]	PTSD, Depression	2	2	2	2	0	0	1	1	1	0
31	Ben-Esra et al. 2011 [57]	PTSD, Depression	2	2	2	2	0	0	1	1	1	0

Low risk of bias 2. High risk of bias 0. Unclear risk of bias 1.

**Table 2 ijerph-17-09369-t002:** Assessment of evidence for the risk of studied outcomes based on Grades of Recommendations, Assessment, Development, and Evaluation framework (GRADE).

Risk	Quality of Study Limitations, ↓	Indirect-Ness of Evidence: ↓	Inconsistency: ↓	Imprecision, Range Confidence Interval Effect Size > 2.0: ↓	Publication Bias, Yes: ↓	Effect Estimate >2.0: ↑ >5.0: ↑↑	Dose-Response Effect: ↑	Residual Confounding: ↑	Overall Certainty (High, Moderate, Low)
PTSD in soldiers after war deployment	no (-) ^1^	no (-)	no (-) ^2^	no (-) 1.83–2.60	no -	no (-) ^3^ 2.18 (1.83–2.60)	no -	no -	moderate
Depression in soldiers after war deployment	no (-) ^4^	no (-)	no (-) ^5^	no (-) 1.06–1.25	no -	no (-) 1.15 (1.06–1.25)	no -	no -	moderate
Depression in workers after exposure to trauma	no (-) ^6^	no (-)	no (-) ^7^	no (-) 1.45–2.15	no -	no (-) 1.77 (1.45–2.15)	no -	no -	moderate
PTSD in workers after exposure to trauma	no (-) ^8^	no (-)	no (-) ^9^	yes ↓ 1.76–5.76	no -	yes ↑↑ ^10^ 3.18 (1.76–5.76)	no -	no -	high

^1^ 9/11 studies had a high risk of bias, and a high risk of bias increased the RR (RR = 2.46; 95% CI 1.41–4.29). However, in low risk of bias studies, the risk is also significantly increased (1.88; 1.61–2.20), and these studies have an important weight (42.18%). ^2^ High I^2^ observed in the overall analysis and in low risk of bias studies. However, the low risk of bias studies had a high number of participants, which increased the value of I^2^. Heterogeneity cannot, therefore, be based solely on values of I^2^. Effect estimates between low risk of bias were similarly high (1.74 and 2.04). The weight of the low-risk studies was important at 42.18%. ^3^ The pooled effect estimate was >2.0, but the high and low risk of bias in studies had pooled effects which differed significantly from each other (high risk RR = 2.46; 95% CI 1.41–4.29; low risk RR = 1.88; 95% CI 1.61–2.20), and the low risk of bias in studies had an effect lower than 2.0. ^4^ 3/5 studies had a high risk of bias, and in studies with a low risk of bias, the RR was significantly increased (high risk RR = 1.60; 95% CI 0.93–2.74; low risk RR = 1.11; 95% CI 1.10–1.13); however, studies with a low risk of bias were statistically significant. The weight of the studies with a low risk of bias was important at 88.97%. ^5^ High heterogeneity was observed in the overall analysis (I^2^ = 75%), but studies with a low risk of bias were homogeneous. ^6^ 7/9 studies had a high risk of bias, and a high risk of bias increased the RR (high risk RR = 2.10; 95% CI 1.90–2.33; low risk RR = 1.44; 95% CI 1.29–1.61); however, the RR in studies with a low risk of bias was also significantly increased. The weight of studies with a low risk of bias is important at 39.90%. ^7^ Moderate heterogeneity was observed in the overall analysis (I^2^ = 64.6%), but studies with a low risk of bias were homogeneous. ^8^ 5/6 studies had a high risk of bias, but in studies with a low risk of bias, the RR was significantly increased (low risk RR = 5.20; 95% CI 4.82–5.61; high risk RR = 2.45; 95% CI 1.36–4.39). The weight of the low-risk studies was important at 30.73%. ^9^ Moderate heterogeneity was observed overall (I^2^ = 65%). ^10^ The pooled effect estimate was 3.18 (greater than 2.0 but less than 5.0), but the studies with a high and low risk of bias had pooled effects which differed significantly from each other (low risk RR = 5.20; 95% CI 4.82–5.61; high risk RR = 2.45; 95% CI 1.36–4.39). The studies with a low risk of bias had an effect greater than 5.0, so the quality level was upgraded twice. “↓” downgrading, “↑” upgrading, “↑↑” double upgrading.

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
