# Peer review of "Occupational Risk for Post-Traumatic Stress Disorder and Trauma-Related Depression: A Systematic Review with Meta-Analysis"

_ijerph, 2020, doi:10.3390/ijerph17249369_

Round 1

Reviewer 1 Report

In the manuscript entitled “Occupational risk for post-traumatic stress disorder and trauma-related depression: a systematic review with meta-analysis,” a round of systematic review and meta-analysis was conducted to understand whether individuals of certain occupations are prone to develop a number of internalizing problems/affective disorders. I believe this manuscript holds great potential and offers an exemplary guideline for future systematic review and meta-analysis. I only have a few suggestions:

  1. The authors were motivated by their prior meta-review that most previous systematic review studies were flawed. It is thus imperative that the authors, in the Discussion section, directly compare the results from those weak systematic review with the current study. In section 4.1, the authors brought up several empirical studies, and this section can be further strengthened when existing systematic review that the authors critiqued is also reviewed.
  2. Figure 1: Please follow PRISMA guideline to specify the stages of identification, screening, eligibility, and included. Furthermore, the “duplicates removed” should be “Records after duplicates removed” to avoid confusion. Please thoroughly revise the flow diagram.
  3. Figure 2 & 3: The authors may want to double check the effect size axis. The numbers do not look right, or some explanation is necessary. The authors can also consider to truncate the axis.
  4. Table 1: There is a misspelling of Darves-Bornoz paper (No. 4)
  5. Table S2: The heading of the third column should be proofread. Also, the “Design_D---Comparison_C…” expression is confusing.

Reviewer 2 Report

Allow me to congratulate for this study which represent an important effort for revising the existing literature on the risks of developing PTSD and depression symptomatology as a consequence of the exposure to occupational traumatic events. From my point of view the study is relevant, well designed and executed and conclusions are strictly supported by the results. I’m grateful for having had the opportunity to have early access to your manuscript, which I hope to see published soon.

I would like to ask you just a question: I have found (you cited it) a previous article published by your team in German (DOI: 10.1055/a-0822-7712) that seems to be very similar to that (excuse me but I cannot read German). Could you, please, explain me what differences are between that article and the actual manuscript?

I am convinced that the study presented in the submitted manuscript is a great study. Although I do not consider myself an expert in statistics, I am confident in that the authors have made a meticulous job and that the results are accurate and soundness. 

In the other hand, I have some reservations about the significance of this kind of meta-analysis. Frequently, and this is the case for the study presented in this manuscript, they merge a so big collection of diverse data that I am not sure that they lead to especially valuable results. In this case, the results show that exposure to traumatic events in an occupational scenario increases the probability of developing symptoms of PTSD and Depression. Maybe the diversity of the sources of data has obscured the results, by the way. I do not feel this is a surprising result. In this sense, I would rather prefer to conduct a more detailed analysis on a smaller set of data (i.e. analyzing each profession separately) that could offer more insight on the risk and protection factors that play a role in developing such symptomatology. However, I am not sure whether this is possible or relevant at all taking into account the available studies. 

Nevertheless, I am aware of the fact that this kind of studies are very prestigious, and, in some circumstances, they might make a crucial contribution to our knowledge in the field concerned. Whereas, sometimes, they do not make a difference though. 

The manuscript herein discussed should not be considered irrelevant at all. Moreover, in my opinion, it is well executed and exposed. This is the reason why I recommended its publication as it is, apart from my concern about its originality, since there is a previous version published in German, as I already noted in my review. 

To conclude, I still believe that it makes sense to publish it despite of my concerns about this type of studies.

Reviewer 3 Report

The authors conducted a systematic review and meta-analysis of occupational risk of PTSD and depression in specific occupational categories, including soldiers, health care workers, police and firefighters, drivers, journalists and war reporters.

Although the topic is important and worthy of further study, this manuscript has numerous problems both in form, content and methodology.

Introduction

A major problem is the lack of any reference to the diagnostic criteria of post-traumatic stress disorder and major depressive disorder. Discussion of differential diagnosis criteria is lacking.

The authors do not clearly and directly specify the hypothesis of the study. We speak in a very general way of verifying “whether there is an increased occupational risk in particular occupational groups or specific occupational trauma for PTSD and depression”. The scientific background based on which there may be an increase in PTSD and depression is not presented, nor in which specific groups or for which specific trauma there is scientific evidence in the literature of increased or possibly increased risk.

The objectives of the study are missing.

Methods

The study inclusion criteria are unclear. The authors indicate two questions at the beginning of the section but do not indicate how they investigated these questions in the studies they screened. It is unclear whether they included 1. patient groups compared to the general population, or 2. trauma patient groups compared to non-trauma patients or 3. groups who developed PTSD versus groups who did not develop PTSD, or all three of these categories (or still other categories based on other criteria).

The statistical methodology followed for the meta-analysis is not clearly specified, nor is it clearly specified what type of data were subjected to meta-analysis.

Results: the authors excluded several studies but the reasons are not clear enough: 25 studies were excluded “mainly due to a lack of response information or due to a response below 10%”. It is unclear if there are other exclusion reasons for these 25 studies.

They also excluded 65 studies for the absence of a comparison group or due to the use of an unsuitable comparison which had similar exposure to that of the study group, but it is not specified what compares to what (depression vs. PTSD? vs. patients with depression? Healthy subjects vs. patients with PTSD? Trauma-exposed versus non-trauma patients? Trauma-exposed patients, who developed PTSD versus patients who did not develop PTSD? Other types of comparison?).

In 16 studies, the exposure was considered not pertinent, but it is unclear as to what should be relevant.

Four studies were excluded based on the outcome, but it is unclear what kind of outcome the authors consider right.

The authors, therefore, included 31 papers, of which only four are considered to be at low risk of bias, which strongly weakens the results of their meta-analysis.

Discussion

Discussion data are reported chaotically. Many different categories of workers are considered and some results reported from some scientific works. The section "strengths and limitations” is too long and should be moved to the end of the discussion.

In the section "PTSD risks due to occupational trauma", the authors report that the risk of PTSD has increased among soldiers after war deployment. However, this is also well known in the clinical community. Subsequently, the authors consider the risk of PTSD in numerous populations put together, but in my opinion, the specificities of the populations in question should lead to consider them separately.

In the section "Depression risks due to occupational trauma" the authors do not clarify what type of depression they refer to (Depressive episode? Major depressive disorder? Depressive episode in bipolar disorder? Other nosographic entities?), and subsequently pose problems of differential diagnosis with adjustments disorders, which is further confusing when one thinks of the methodology used and data presented.

Finally, the conclusions focus more on the risk of PTSD and depression in soldiers exposed to trauma in war, which is already widely established in the literature.

Reviewer 4 Report

This is a very interesting and important contribution to the field of occupational trauma and stress research.  This is one of the better manuscripts I have reviewed in a long time.  I have very few comments for the authors.

  1. Minor formatting:  There are a few places where the fonts and formatting of the paper were different, (see lines 515-522).  
  2. Methodological: There are some related articles that seem to not have been included in the 141 screened articles.  One in particular is Chin and Zeber (2020)Mental health outcomes among military service members after severe injury in combat and TBI. Mil Med 185(5-6); e711–e718, but this might have been too recently published to meet inclusion (reviewer is not an author). Others include (1) Sandweiss et al (2011)Preinjury psychiatric status, injury severity, and postdeployment posttraumatic stress disorder. Arch Gen Psychiatry 68(5): 496-504, (2) Porter B, et al. (2018) Measuring aggregated and specific combat exposures: associations between combat exposure measures and posttraumatic stress disorder, depression, and alcohol-related problems. J Trauma Stress; 31:296–306, and (3) Armenta R et al (2018). Factors associated with persistent posttraumatic stress disorder among US Military Service Members and Veterans (Open Access Publisher’s Version). BMC Psychiatry; 18:48 (reviewer is not an author of any of these articles).  None of these were included in the 141 screened articles, yet seem like they should have been included.  These are just a couple, and there could be others. This makes me wonder why these articles were not identified.  This should be addressed. 

Round 2

Reviewer 3 Report

The manuscript still has different issues both in the scientific methods and discussion of the study results. The authors did not reply on different problems already highlighted in the previous review.

Introduction: the issue of the lack of scientific background and references for post-traumatic stress disorder and major depressive disorder persists. The authors, instead of providing such background in the introduction, have added some aspects into the discussion. However, this addition is in turn problematic and inconsistent, as they highlighted methodological inconsistencies of the included studies and the absence of specialist diagnoses in "some singular studies". The sentence "A gold standard for the evaluation of PTSD and depression has not been agreed upon yet" is too generic and out of context.

The authors do not clearly and directly specify the hypothesis of the study. They generally wrote about verifying "Whether there is an Increased occupational risk in Particular occupational groups or specific occupational injury for PTSD and depression". The scientific background based on which there may be an increase in PTSD and depression could be better presented (it is important to underline in which specific groups or for which specific trauma there is evidence of increased or possibly increased risk).

The authors moved inclusion and exclusion criteria from a supplementary table to table 1, and it is still unclear whether they included 1. patient groups compared to the general population, or 2. trauma patient groups compared to non-trauma patients or 3. groups who developed PTSD versus groups who did not develop PTSD, or all three of these categories (or still other categories based on other criteria). Comparison conditions included in their meta-analysis are lacking. An additional table indicating all the comparison conditions could be useful.

The objectives of the study are still missing.

The process of inclusion and exclusion of studies is still unclear. In Figure 1, the process leading to the inclusion of 23 studies is not clear. An additional Flow-chart figure dedicated only to the meta-analysis process would be useful.

The discussion is still inconsistent. The "strengths and limitations section" could be shortened and moved at the end of the discussion. It could be useful to discuss the results obtained and then consider the self-referential aspects. It would be important to discuss the data describing the issues related to individual categories of workers or to more homogeneous groups that include subjects that do similar types of works.

The issue relating to the adjustment disorder remains in the discussion. The authors should consider this diagnosis only concerning any studies included in the review in which the sample consists of subjects suffering from adaptation disorder.

Conclusions remain weak, and the major findings of the study should be presented more clearly and directly.

Finally, the English language should be reviewed.
